# Importance of between and within Subject Variability in Extracellular Vesicle Abundance and Cargo when Performing Biomarker Analyses

**DOI:** 10.3390/cells10030485

**Published:** 2021-02-24

**Authors:** Lauren A. Newman, Alia Fahmy, Michael J. Sorich, Oliver G. Best, Andrew Rowland, Zivile Useckaite

**Affiliations:** College of Medicine and Public Health, Flinders University, Adelaide, SA 5042, Australia; lauren.newman@flinders.edu.au (L.A.N.); alia.fahmy@flinders.edu.au (A.F.); michael.sorich@flinders.edu.au (M.J.S.); giles.best@flinders.edu.au (O.G.B.); andrew.rowland@flinders.edu.au (A.R.)

**Keywords:** extracellular vesicles, diurnal variability, biomarkers, liquid biopsy, liver specific

## Abstract

Small extracellular vesicles (sEV) have emerged as a potential rich source of biomarkers in human blood and present the intriguing potential for a ‘liquid biopsy’ to track disease and the effectiveness of interventions. Recently, we have further demonstrated the potential for EV derived biomarkers to account for variability in drug exposure. This study sought to evaluate the variability in abundance and cargo of global and liver-specific circulating sEV, within (diurnal) and between individuals in a cohort of healthy subjects (*n* = 10). We present normal ranges for EV concentration and size and expression of generic EV protein markers and the liver-specific asialoglycoprotein receptor 1 (ASGR1) in samples collected in the morning and afternoon. EV abundance and cargo was generally not affected by fasting, except CD9 which exhibited a statistically significant increase (*p* = 0.018). Diurnal variability was observed in the expression of CD81 and ASGR1, which significantly decreased (*p* = 0.011) and increased (*p* = 0.009), respectively. These results have potential implications for study sampling protocols and normalisation of biomarker data when considering the expression of sEV derived cargo as a biomarker strategy. Specifically, the novel finding that liver-specific EVs exhibit diurnal variability in healthy subjects should have broad implications in the study of drug metabolism and development of minimally invasive biomarkers for liver disease.

## 1. Introduction

Small extracellular vesicles (sEVs), in particular exosomes, have emerged as a rich source of circulating biomarkers with applications including tracking variability in disease, intervention efficacy, and drug exposure [1,2,3]. sEVs are heterogeneous membrane encapsulated particles of less than 150 nm in diameter that are secreted into the blood and other biofluids by virtually all cell types [4]. The sEV class comprises multiple specific EV types including exosomes (classical and non-classical), arrestin-domain-containing protein 1-mediated microvesicles (ARMM), small apoptotic EV, and small autophagic EV; however, multiple larger EV classes also exist [5]. These vesicles contain a complement of nucleic acid (microRNA (miRNA), mRNA, and non-coding RNA), protein, and small molecule cargo that are derived from their cell of origin [6]. For the purpose of this study, we refer to this heterogenous population of isolated vesicles as EVs.

Some EV cargo is explicitly packaged through defined pathways that are specific to a particular EV type while other cargo is indiscriminately incorporated as a by-product from the cellular milieu [6]. Accordingly, the composition of EV cargo depends on EV type and the cell of origin. Differences in cargo between EV types have been extensively studied and several markers have been proposed to discriminate sEV based on their type (e.g., CD9, CD63, CD81, TSG101, and Calnexin). EVs derived from the same cell have been shown to vary in molecular composition [7]; yet, the degree of normal variability in the abundance of circulating sEV and their cargo remains poorly defined [8]. In order to robustly define thresholds of accuracy, precision and sensitivity for an EV derived biomarker, it is essential to understand the normal degree of variability in circulating EV and to understand patterns (e.g., circadian) associated with EV abundance.

Of growing interest is the understanding of how EV derived from a specific cellular or subcellular location may be applied to gain even greater understanding of organ function. There are a number of studies that have defined protocols for the isolation of tissue- and organelle- specific EVs based on selective surface proteins [9,10,11,12]. By way of example, liver-derived EVs can be selectively captured via the hepatocyte-specific surface protein asialoglycoprotein receptor 1 (ASGR1), and may be of great value to the study of drug metabolism [2] or liver diseases [4]. In order to robustly define abnormal expression, these applications require an understanding of the normal range of expression between individuals, typical variability in expression within an individual, and the contribution of different tissues to the global EV pool in circulation. This study sought to evaluate the variability in circulating global and liver specific sEV abundance and cargo, and to define patterns of variability, within (diurnal) and between individuals, that have the potential to confound sEV derived biomarker analyses. The experimental design for the study is summarised in Figure 1.

## 2. Materials and Methods

### 2.1. Study Cohort

EVV is a single-centre, open-label, single-sequence biomarker study involving a cohort of healthy males and females aged 18 to 65 years. Characteristics of the study cohort are detailed in Table 1. The study protocol was approved by the Southern Adelaide Clinical Human Research Ethics Committee (SAHREC 261.18), and written informed consent was obtained from each participant prior to study enrolment. The study was conducted according to the principles stated in the Declaration of Helsinki, and was compliant with CPMP/ICH/135/95 GCP standards.

### 2.2. Collection of Blood/Serum

Morning samples were collected between 9:00 and 9:30 am, while afternoon samples were collected between 3:00 and 3:30 pm. Participants presented fed for blood collection on day 1 and after an overnight fast on day 3. The participants were free to consume their regular meals of choice for the duration of the study when not required to be fasted. Eight millilitres of whole blood was collected into Z Serum Sep Clot Activator tubes (Greiner Bio-One, Frickenhausen, Germany) using a 21-gauge Vacuette Safety Blood Collection sealed vacuum device (Greiner Bio-One, Frickenhausen, Germany). To ensure sample quality the device was primed by collecting a 5 mL ‘discard’ tube immediately prior to sample collection. Serum was isolated from whole blood within 60 min of sample collection by two cycles of centrifugation at 2500× *g* for 15 min at 4 °C.

### 2.3. Extracellular Vesicle Isolation

Global EVs were isolated from serum by size exclusion chromatography (SEC) performed using 2 mL qEV70 columns (Izon Science, Christchurch, New Zealand). SEC methods have been shown to effectively separate EVs from lipoproteins and other contaminants in serum, and is attractive for clinical biomarker applications due to its scalability and efficiency [13,14,15]. The use of 2 mL columns ensured all downstream analyses came from the same isolation. Prior to EV isolation, SEC columns were conditioned by washing with 10 mL of freshly 0.2 µm filtered phosphate-buffered saline (PBS). Thawed serum (2 mL) was added to the sample reservoir and EVs were eluted in PBS, which was added to the sample reservoir as the last of the serum entered the column. For the duration of the EV isolation, the volume of PBS in the reservoir was kept below 2 mL. The initial six fractions of flow-through were discarded, EVs were collected in the pooled fractions 7 to 11 using Protein LoBind tubes (Eppendorf, South Pacific, Australia). Resulting pooled EV fractions were mixed by gentle inversion 8 to 10 times and concentrated using pre-conditioned 30 kDa Amicon Ultra-4 centrifuge filters (Millipore-Sigma, Bedford, MA, USA) to a final volume of 400 µL in PBS. EV samples were aliquoted to avoid freeze–thaw and stored at −80 °C until analysed or processed further.

### 2.4. Human Liver Microsome Preparation

Pooled human liver microsomes (HLMs) were prepared by mixing equal protein amounts of microsomes from five human livers (H7, 44-year-old woman; H10, 67-year-old woman; H12, 66-year-old man; H29, 45-year-old man; and H40, 54-year-old woman) obtained from the human liver “bank” of the Department of Clinical Pharmacology (Flinders University, Adelaide, SA, Australia). Approval for the use of human liver tissue in xenobiotic metabolism studies was obtained from both the Clinical Investigation Committee of Flinders Medical Centre and from the donors’ next of kin. HLMs were prepared by differential centrifugation, as described by Bowalgaha et. al. [16].

### 2.5. Nanoparticle Tracking Analysis

Nanoparticle tracking analysis (NTA) was performed to determine global particle abundance and size distribution using the NanoSight NS300 (Malvern Panalytical, Malvern, United Kingdom, Software Version 3.4). Samples were diluted between 1:1000 and 1:5000 using freshly 0.2 µm filtered PBS; five 60-s videos were captured and analysed under constant flow conditions (flow rate 50) using NTA 3.4 software.

### 2.6. Transmission Electron Microscopy (TEM)

Samples were prepared adapting a previously published protocol, with some modification [17]. Briefly, Ted-Pella B 300M carbon-coated grids (Ted-Pella, Redding, CA, USA) were cleaned and hydrophilized using plasma glow discharge for 15 s (Gatan SOLARUS Advanced Plasma Cleaning System, Gatan, Inc., Pleasanton, CA, USA) prior to use. Five μL of sample in 0.2-µm-filtered PBS was placed on carbon-coated grids for 5 min. Carbon grids were washed once (15 s) at room temperature (RT) with 0.2 µm filtered PBS and were contrasted with 2% uranyl acetate (3 min, RT), washed once, and examined by FEI TECNAI Spirit G2 TEM (Thermo Fisher Scientific, Waltham, MA, USA) operated at 100 kV. TEM images were acquired at 30,000× and 68,000× (Appendix A).

### 2.7. Micro BCA Protein Quantification

A micro bicinchoninic acid (BCA) reagent kit (ThermoFisher Scientific, Waltham, MA, USA), with a bovine serum albumin (BSA) standard curve (0–200 μg/mL), was used to determine total protein in samples, as per the manufacturer’s recommendations. EVs were lysed by addition of RIPA buffer 1:1 and incubation on ice for 25 min, followed by centrifugation at 10,000× *g* for 10 min at 4 °C. Working reagent (WR) was prepared using MA:MB:MC at a 25:24:1 ratio, respectively. EV containing samples were diluted between 1:20–1:100 in 0.2-µm-filtered PBS; 150 μL of the WR was mixed with either 150 μL of the BSA standard or 150 μL of sample in duplicate using a 96-well plate. The plate was covered and incubated at 37 °C for 2 h. Absorbance was measured at 562 nm using a SpectraMax plate reader (Molecular Devices, San Jose, CA, USA).

### 2.8. Trypsin Digest

EV protein was prepared for LCMS analysis by trypsin digestion. EVs (50 µL) diluted up to 100 µL in PBS were vortexed for 10 min using a MixMate (Eppendorf, South Pacific, Australia) and subject to three freeze–thaw cycles to cause lysis. Dithiothreitol (12.5 mM) and 50 µL of ammonium bicarbonate (pH 7.8) were added to samples and incubated for 90 min at 60 °C. Samples were cooled to room temperature, then incubated with iodoacetamide (23.5 mM) for 60 min at 37 °C. The samples were then incubated for 18 h at 37 °C with Trypsin Gold (Madison, WI, USA) at a Trypsin to protein ratio of 1:40. Trypsin digestion was terminated by addition of 20 µL formic acid and centrifuged at 16,000× *g* for 10 min at 4 °C. Supernatant (100 µL) was extracted and stable isotope labelled (SIL) peptide standards (New England Peptide, Gardner, MA, USA) were added at 2.5 nM concentration. A 5 µL aliquot was injected for analysis by LC-MS/MS. Serum (diluted 1:10,000) and HLM were digested in the same conditions and run as positive controls.

### 2.9. Liquid Chromatography Mass Spectrometry (LCMS)

Chromatographic separation of analytes was performed on an Agilent Advance Bio Peptide Map column (100 mm × 2.1 mm, 2.7 µm) using an Agilent 1290 Infinity II liquid chromatography system (Agilent, Santa Clara, CA, USA). The temperature of the sample and column compartments was maintained at 4 and 30 °C, respectively. A panel of analytes comprising the EV makers CD9, CD63, CD81, and contaminants calnexin V and albumin, were separated by gradient elution with a flow rate of 0.2 mL/min. The mobile phase consisted of 0.1% formic acid in water (mobile phase A) and in acetonitrile (mobile phase B) held in a proportion of 90% A and 10% B for the first 2 min. The proportion of mobile phase B was then increased linearly to 60% over 13 min before returning to 10% over the next 4 min and held for a further minute to re-equilibrate. Total run time was 20 min. EV marker TSG101 was run independently using isocratic elution at a flow rate of 0.2 mL/min. Mobile phase A was held at 80% and total run time was 6 min. The liver-specific EV protein marker ASGR1, also run independently, was separated by gradient elution at 0.2 mL/min. Mobile phase B was increased linearly from 10% to 40% over 8 min then returned to 10% over 1.4 min. The column was re-equilibrated with a total run time of 10 min. Column eluant was monitored by mass spectrometry using an Agilent 6495B Triple Quadrupole mass spectrometer operating in positive electron spray (ESI^+^) mode. Multiple reaction monitoring (MRM) was performed with one quantifier and one qualifier ion transition for each peptide. Identities of endogenous peptides were confirmed by comparing retention time and quantifier/qualifier transition ratios to respective SIL peptide standards. Analyte peptide sequences are given in Appendix A.

### 2.10. Nano Flow Cytometry (nFC)

Flow cytometry analysis was performed on the Beckman Coulter CytoFLEX S Flow Cytometer (Beckman Coulter, Brea, CA, USA) as previously described [18]. Briefly, for daily calibration of the flow cytometer, Megamix FSC & SSC Plus, BioCytex fluorescent beads (BioCytex, Marseille, France) were used in sizes of 100, 160, 200, 240, 300, 500, and 900 nm. The gating strategy is described in Appendix A. The VSSC and SSC threshold was set as the trigger channel below the 0.1-µm bead population. A rectangular gate was set on the VSSC-H log × BSSC-H log cytogram containing the 100 nm and 240 nm bead populations and defined as ‘100 nm–240 nm Megamix gate’ followed by a “stable time gate” set on the time histogram, in order to identify the microparticle region. To avoid swarm effects each was serially diluted from 1:2 to 1:500 and measured with a flow rate of 10 µL/min prior to antibody labelling. EVs were labelled with 0.05 µL of anti-CD63-AlexaFluor488 (Invitrogen/Thermo Fisher Scientific, Waltham, MA, USA, Cat.MA5-18149) or anti-CD9-BV405, (R&D Systems, Minneapolis, MN, USA, Cat.FAB1880V), anti-CD81-Alexa700 (Biolegend, San Diego, CA, USA, Cat.349518), anti ASGR1-BV421 (BD Biosciences, San Jose, CA, USA, Cat.74269) in 100 µL PBS for 30 min on ice in the dark. To avoid false positive event measurement, all antibodies used were run in PBS alone to ensure the absence of antibody aggregates and non-specific binding to the particles in PBS. To avoid carry-over effects between each sample measurement, a washing step was performed with filtered PBS for 1 min at an increased flow rate of 60 µL/min. EV lysis was performed by incubating PBS-diluted EVs in 0.05% Triton™ X-100 for 30 min at room temperature.

Based on recommendations in the MIFlowCyt-EV guidelines [19], buffer only (PBS) control, buffer with antibodies, unstained controls (EVs in PBS), and stained EVs were run under the same settings (Appendix A).

### 2.11. Statistical Analysis

Statistical analysis was performed using GraphPad Prism software (San Diego, CA, USA, version 9.0). The D’Agostino-Pearson omnibus K2 test was used to assess normality and log transformation was applied to NTA and mass spectrometry data. All variables passed normality and lognormality tests, so parametric tests were applied, except in the case of total protein concentration, to which Wilcoxon test was used instead. Data was presented as mean ± 95% confidence interval and range. Statistical comparisons were performed between different time points (AM and PM, fed and fast) using paired *t*-tests and between independent groups (sex) using one-way ANOVA. Statistical significance was set at 0.05.

### 2.12. EV-TRACK

We have submitted all relevant data of our experiments to the EV-TRACK knowledgebase (EV-TRACK ID: EV210044) [20].

## 3. Results

### 3.1. Purity Assessment of EV Isolations from Serum

In order to assess the purity of EV isolations, a few samples were selected at random and imaged by TEM to evaluate the background, composition, and EV structure. Representative images from TEM analysis (Figure 2, Appendix A) indicated that EV populations obtained by qEV70 SEC columns had limited non-vesicular contamination and the majority of EVs were 40–140 nm in size. EVs were round and appeared structurally intact.

Adhering to EV TRACK transparent reporting platform recommendation, assessment of non-EV-enriched proteins was performed in all EV samples to evaluate EV sample purity and potential non-vesicular contamination [20]. Calnexin V and albumin levels were measured by LC-MS. Total protein matched serum and HLM samples were used as positive controls for albumin and Calnexin V, respectively. The abundance of negative markers in EV samples is presented as a mean percentage of expression ± SD relative to HLM and serum for the respective markers. Compared to HLM, minimal expression of Calnexin V was detected in EV samples (0.52 ± 0.40%). Similarly, albumin expression in EVs was minimal compared to the serum control (0.95 ± 0.32%).

### 3.2. Normal Variability

Normal variability between individuals was assessed with respect to EV characteristics and the abundance of EV-associated protein markers on study day 1. Nanoparticle tracking analysis (NTA) was employed to determine size and concentration of particles in EV isolates (Figure 3A,B). The mean (±range) was 2.82 × 10^11^ (3.02 × 10^10^–1.26 × 10^12^) and 2.40 × 10^11^ (4.37 × 10^10^–1.02 × 10^12^) particles/mL for AM and PM samples, respectively (Table 2). Mode size of particles was 83.0 (64.9–99.3) nm in the morning and 84.7 (76.0–92.9) nm in the afternoon. Total protein in lysed EV samples was determined by microBCA assay and varied widely between participants in the morning and afternoon (Figure 3C). Mean (±range) concentration in respective AM and PM samples was 2773 (26.4–2799.0) µg/mL and 1318 (27.9–1345.9) µg/mL (Table 2). Quantification of EV protein, particle concentration, and size in the present study was consistent with previously reported ranges for EVs isolated by qEV from human serum [14].

Markers were selected based on the Minimal Information for Studies of Extracellular Vesicles (MISEV) guidelines for confirming the presence of EV-enriched proteins and include those derived from the plasma membrane, endosomal pathway, and cytosol [21]. The panel was comprised of tetraspanins CD9, CD63 and CD81, and the cytosolic protein tumour susceptibility gene 101 (TSG101), as well as the hepatocyte-specific surface protein asialoglycoprotein receptor 1 (ASGR1), which is known to be expressed on EVs derived from this cell type [4,22]. The expression of protein markers was quantified by LCMS and assessed for variability between subjects in AM and PM EVs (Figure 3D–H). Generic EV markers (tetraspanins and TSG101) showed relatively low variation between subjects, except for CD63 which was significantly more variable in both the morning and afternoon. The relative response values ranged from 2.6 to 1158.8 in AM samples and 1.3 to 4954.5 in PM samples (Table 2). The liver-specific EV protein ASGR1 exhibited similarly high variability, but only in the afternoon, as the range of values exceeded 16 times that of the morning EV samples. In the context of biomarker applications, understanding these differences in marker expression within and between individuals may aid the optimisation of sampling protocols.

### 3.3. Effect of Fasting

Quantification by NTA of serum EVs collected from participants in fed and fasted states, revealed respective mean (±95% CI) particle concentrations of 2.82 × 10^11^ (1.07 × 10^11^–7.24 × 10^11^) and 2.09 × 10^11^ (6.61 × 10^10^–6.46 × 10^11^) particles/mL. No significant differences were detected by paired statistical tests (Table 3, Figure 4A,B). The mode particle size was slightly higher after fasting at 92.0 (80.4–105.4) nm compared to 83.0 (75.2–91.8) nm from fed individuals, but this difference did not reach statistical significance (*p* = 0.093).

Total protein concentration and the abundance of EV protein markers in fed and fasted states were also compared (Table 3, Figure 4C–H). Mean protein concentration (±95% CI) in respective fed and fasted states was 345.7 (104.2–1161.5) µg/mL and 822.2 (619.4–1094.0) µg/mL. While the difference between groups was not significant, the inter-individual variability was notably greater in fed samples (Figure 4C). Assuming constant stoichiometry, it follows that a lack of difference in particle concentration should be accompanied by no change in EV protein marker abundance. This was true of all markers except for CD9, which exhibited a statistically significant increase in fasted individuals (*p* = 0.018). Similarly to observations of variability in CD63 abundance throughout the day (Figure 3F), a wide range was exhibited in fed and fasted states (Table 3, Figure 4F).

### 3.4. Diurnal Variability

In order to establish potential patterns of EV variability in healthy subjects, the analysis of EVs collected on study days 1 and 3 were pooled and compared between morning and afternoon. EV abundance and size as measured by NTA was consistent between the two time points (Figure 5A,B). Mean (±95% CI) particle count in respective AM and PM samples was 2.29 × 10^11^ (1.23 × 10^11^–4.79 × 10^11^) and 2.40 × 10^11^ (1.41 × 10^11^–3.89 × 10^11^) particles/mL (Table 4). Mode size of particles was 87.1 (81.3–95.5) nm in AM and 83.2 (77.6–87.1) nm in PM samples.

Analysis of EV protein and abundance of generic markers revealed no difference in total protein concentration or response for CD9, CD63, and TSG101 at different times of the day (Table 4, Figure 5C,E,F,H). Interestingly, however, CD81 was significantly lower in the afternoon compared to the morning (*p* = 0.011) (Figure 5G). With no concomitant change in particle number, this result may suggest a lower CD81 abundance per vesicle or a decrease in the proportion of CD81^+^ EVs. Additionally, a significant increase in ASGR1 response was observed from AM to PM samples (*p* = 0.009), suggesting that the proportional contribution of the liver to the circulating global EV pool was greater in the afternoon (Figure 5D).

### 3.5. Effect of Sex

The impact of sex was next explored as a potential source of variability in serum EV abundance and composition. Pooled analysis was performed for EVs collected on study days 1 and 3 in the morning and afternoon and compared between female (*n* = 10) and male (*n* = 10) healthy subjects. In this cohort, EV concentration in AM samples as determined by NTA was more than 10 times greater in males compared to females (*p* < 0.0001) (Figure 6A). Mean (±95% CI) particle count in female and male cohorts was 7.41 × 10^10^ (3.47 × 10^10^–1.58 × 10^11^) and 7.76 × 10^11^ (5.62 × 10^11^–1.10 × 10^12^) particles/mL (Table 5). This difference was less substantial in PM samples, at 1.10 × 10^11^ (6.03 × 10^10^–1.95 × 10^11^) particles/mL in females and 5.01 × 10^11^ (3.24 × 10^11^–7.94 × 10^11^) particles/mL in males, but retained statistical significance (*p* = 0.0002). Meanwhile, mode size of particles did not vary with sex in either the morning or afternoon (Table 5, Figure 6B). In AM EVs, mean (±95% CI) protein concentration was 330.4 (154.5–706.3) µg/mL in females and 865.0 (335.0–2238.7) µg/mL in males. This significantly higher mean concentration in male subjects (*p* = 0.037) did not persist into the afternoon (Table 5, Figure 6C).

The mean abundance of EV protein markers quantified by LC-MS showed no variations according to sex (Figure 6D–H, Table 5). Interestingly, the previously described diurnal pattern of ASGR1 response was exhibited in female and male subjects alike and no significant differences were observed between the two cohorts (Figure 6D). CD63 response also showed the same degree of variability between subjects as observed for the combined cohort, indicating that this finding was not singly influenced by either sex (Figure 6F).

### 3.6. Single EV Analysis by Nano Flow Cytometry

In addition to EV sample analysis by LC-MS, the presence of EV markers CD9, CD63, CD81, and ASGR1 was confirmed by nano flow cytometry (Appendix A). A Cytoflex S instrument was used to analyse surface EV protein markers on intact, individual EVs, providing an additional insight into the EVV study using an alternative platform.

Normal ranges of EVs positive for CD9, CD63, CD81, and ASGR1 and quantification by mean fluorescence intensity (MFI) was assessed for normal variability between the individuals and diurnal variability on day 1 (Appendix A). No significant differences were observed for tetraspanin EV protein markers or for the liver-specific EV protein ASGR1, however, ASGR1 levels were noticeably the most variable between individuals in both the AM and the PM, being more pronounced in the PM samples (Appendix A). MFI values ranged from 241.4 to 582.0 in AM samples and 235.1 to 884.0 in PM samples. Importantly, the range of observed PM sample values was 1.6 times greater than that of morning EV samples (Appendix A). Difference in MFI from AM to PM for CD9^+^, CD63^+^, and CD81^+^ EVs was 3.5-, 1.5-, and 1.2-fold, respectively. Single EV analysis revealed a greater inter-individual variability in fed samples for ASGR1, CD63 and CD81 (Appendix A), and CD9 and CD81 levels were significantly different (*p* = 0.049 and *p* = 0.002, respectively) between fed and fasted samples.

Differences in diurnal variability were observed for ASGR1 and CD81 (Appendix A). ASGR1 was noticeably the most variable between individuals in both the morning and afternoon and the difference was more pronounced in the afternoon (Appendix A). MFI values ranged from 201.4 to 582.0 in AM samples and from 235.1 to 884.0 in PM samples. MFI values for ASGR1 were higher in PM samples (*p* = 0.002). Importantly, the range of observed PM sample values was 1.7 times greater that of morning EV samples. The difference in MFI between CD81 positive EVs in AM and PM samples was 1.1-fold with MFI range between 6.4 and 137.0 in the AM samples and between 9.3 and 155.8 in the PM samples (*p* = 0.036).

Differences in MFI values of CD81 positive EV populations were observed between males and females in both morning (*p* = 0.038) and afternoon samples (*p* = 0.028) (Appendix A). Interestingly, inter-individual variability was greater in females than males in the AM and PM samples with MFI values ranging between 16.4–137.0 in female AM and 6.4–97.5 in male AM samples. Similarly, MFI values in the range between 31.6–155.8 was observed in female PM samples and between 9.3–96.0 in the male PM samples.

## 4. Discussion

Here, we report for the first time a diurnal pattern of expression for the liver-specific EV marker ASGR1 in healthy human serum. We observed greater abundance and wider variability between subjects in samples collected in the afternoon, indicating that the contribution of the liver to the global pool of circulating EVs changes throughout the day. Importantly, this pattern was consistent in males and females. Notably, in contrast to the observed diurnal variability in expression of ASGR1^+^ EVs, no difference in expression was observed between fed and fasted states. This observation indicates that the diurnal variability in expression of ASGR1^+^ EVs is not a post-prandial phenomenon. Data presented here indicate that accounting for diurnal variability in EV expression may be particularly important for the analysis of liver-specific biomarkers. Liver-specific EV markers are of relevance to the use of EVs in the study of drug metabolism [2,23] and non-alcoholic fatty liver disease (NAFLD) [4]. While not addressed specifically in the current study, these data also raise the possibility that diurnal variability may cofound the analysis of EV markers originating from other tissues and have broader implications for the design of sampling protocols for other tissue-specific markers that may vary in a similar manner.

The potential for circulating EVs and their molecular cargo to be applied as minimally invasive biomarkers is increasingly recognised for the diagnosis of a variety of conditions or tracking individual responses to pharmacological interventions [24]. However, a comprehensive understanding of how these circulating markers fluctuate in normal physiology is currently lacking. Thus, the present study involved the analysis of EVs isolated from the serum of healthy subjects collected at multiple time points. We reported ranges, reflective of the normal variability between individuals, for particle size and concentration of EV isolates, total EV protein concentration and the abundance of EV-associated protein markers. In establishing normal ranges for the characteristics and composition of EVs, investigators may better define the thresholds for disease-associated changes, thereby strengthening the foundations for diagnostic or prognostic applications.

As EVs and their molecular cargo are involved in numerous functions vital for homeostasis, their biogenesis and composition are readily altered by different cellular and extracellular stimuli, including changes in nutrient availability [25]. Circulating biomarkers may fluctuate in response to feeding or alternatively, their quantification may be confounded by natural variation in unrelated blood parameters, especially triglyceride and lipoprotein levels [26]. To explore the effect of prandial state on the characteristics of EVs and abundance of associated protein markers in healthy individuals, serum EVs were compared with and without an overnight fast and it was found that particle number and size was not altered by fasting. These data contrast with prior reports of circulating EVs post-prandially. Mørk et. al. [27] found that food intake resulted in a 61% increase in particle count and a significantly greater median size. However, that study and later work by Jamaly et. al. [26], reported strong correlations between particle count and plasma triglyceride concentrations after feeding, suggesting the similarly sized lipoprotein particles interfered with measurements. Blood serum EVs are unavoidably co-isolated with a range of non-vesicular materials such as protein aggregates and lipoproteins [28,29]. Neither of the aforementioned studies investigated EV purity. The latter of which isolated vesicles by ultracentrifugation, which is known to be more susceptible to lipoprotein contamination [18], rather than size exclusion chromatography (qEV).

It is important to note that NTA lacks the capacity to distinguish vesicles from other particles of comparable size, such as contaminating protein aggregates or lipoproteins [26]. Our assessments of EV purity by TEM and non-EV protein controls indicated highly pure vesicle preparations. Thus, effective removal of lipoproteins from both fed and fasted EV samples may account for differences seen in previous reports and mitigate the need for fasting in biomarker testing.

For the most part, the abundance of EV-associated protein markers, reported here, agrees with a recent study [30] that employed a flow cytometry approach with specific detection of EV protein markers, including the tetraspanins (CD9, CD63, and CD81). While a significant increase in CD9 abundance was observed here, Mørk et. al. [30] found no changes between prandial states. It should be noted, however, that flow cytometry methods are limited to vesicles > 100 nm in diameter, while LCMS permits analysis of all digested vesicular proteins in the samples [31].

The effect of circadian rhythm is well appreciated across numerous physiological systems and presents in circulation as oscillations in haematological parameters, blood and immune cell activation, and expression of surface markers [28]. This study sought to ascertain whether the characteristics of EVs and expression of associated markers exhibit diurnal variability. Particle size and concentration of EV isolates collected in the morning and afternoon did not vary, while the abundance of CD81 decreased and ASGR1 increased significantly.

While there is very limited commentary on the presence of diurnal variation in EV particle number and size over the course of a day, one study utilising nFC, reported an upward trend in EV size as well as a wider range in evening samples compared to the morning [31]. As previously mentioned, these disparate conclusions may be attributed to the capacity for NTA to quantify particles in a size range below that of flow cytometry. Taken together, the results presented here do not support fluctuations in the number or size of EVs diurnally, but point to potential changes in their molecular composition. Accordingly, attention should be given to the time of day at which EVs are sampled to reduce the effects of intra- and inter-individual variability on the sensitivity of biomarker analyses.

Lastly, the participants’ sex was explored as a potential covariate associated with variability in EVs from healthy subjects. NTA analysis revealed that EV samples taken in the morning from males had more than 10 times greater particle concentration than those from females. This difference persisted, albeit at to a lesser extent, in the afternoon. Notably, the stark difference in particle concentration between sexes was not accompanied by greater levels of generic EV markers in male subjects. Sex differences have been observed in prior studies using flow cytometry, whereby plasma-derived phosphatidylserine and other microvesicle markers [32], and urinary CD63^+^ EV levels were each higher in women [33]. Recently, however, an NTA analysis of plasma EVs isolated by precipitation found no difference in particle count between males and females [34]. The pool of circulating EVs is contributed to by numerous cell types, but is largely made up of those released by platelets [8]. Though currently unclear, particle number in male subjects in our study may have been influenced by undefined factors, such as diet, physical activity, and immune activation that prompted the release of particular EV subpopulations bearing cell-type specific markers [28,35,36]. While interesting as an observation, in the absence of controlling for other sources of variability, the data presented here demonstrating differences between sexes in particle abundance should be interpreted with caution as other factors such as exercise may have confounded the results in this small cohort.

Flow cytometry is an appealing tool that lends itself to the analysis of individual protein markers on the surface of intact EVs [18]. Based on our results, it is evident that the data obtained by flow cytometry is not directly comparable to that from LCMS and has some limitations. A large proportion of EVs cannot be included in the quantification due to their size and the limit of detection of the flow cytometer. Conventional flow cytometers are capable of detecting EVs of 100 nm in diameter or greater, thus excluding all smaller EVs [37]. In this study, EVs between 100 and 900 nm were analysed. While observed diurnal variations were not consistent between study participants, every participant showed some level of diurnal variation. This phenomenon was previously shown, however, and studies involving a larger number of participants are required to fully appreciate diurnal changes [31].

Moreover, further detailing the daily time course of changes in EVs and their cargo may facilitate the tracking of therapeutic interventions. Such objectives would be serviced by longitudinal studies, testing in more frequent intervals of the circadian clock, and repeating measures across multiple days. Assessment of normal variability in healthy subjects might also be extended to include the effects of race and other demographic or clinical features.

## 5. Conclusions

Circulating EVs have immeasurable potential to be utilised as biomarkers. The value of this diagnostic and prognostic tool with respect to sensitivity and specificity, however, requires a fundamental understanding of the differences that naturally exist in the healthy population. The findings of this study should, therefore, inform EV sampling and may be of particular importance in the context of liver-specific EV-derived biomarkers.

## Figures and Tables

**Figure 1 cells-10-00485-f001:**
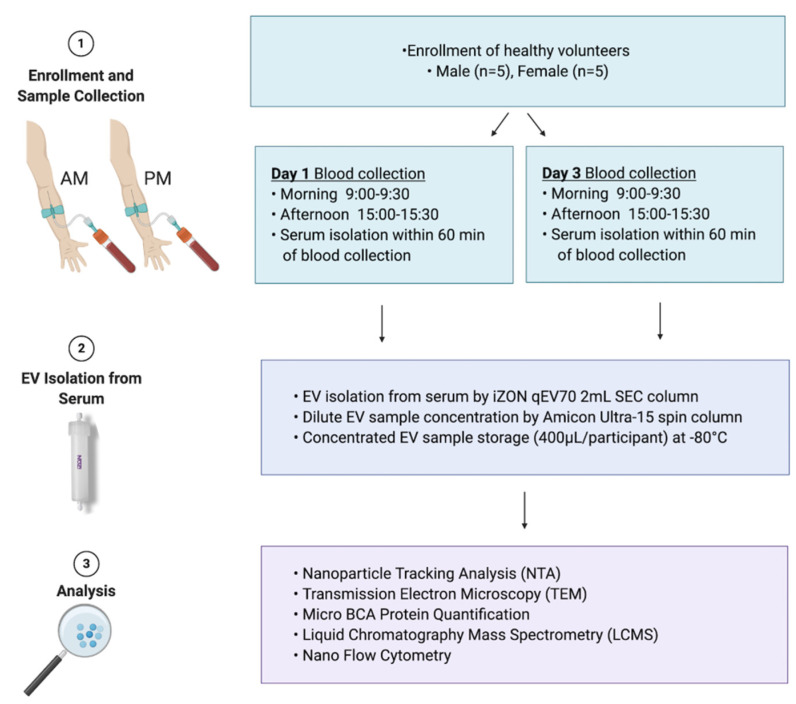
Study design. A cohort of healthy males and females aged 18 to 65 years were recruited (*n* = 10) for extracellular vesicle variability study (EVV). Blood was collected in the morning (AM) and the afternoon (PM) on day 1 and day 3 of the study. Participants were fed on day 1 and fasted on day 3 for morning blood collection. Serum was isolated from whole blood and used for extracellular vesicles (EV) isolation. EVs were isolated using qEV70 2 mL SEC column, concentrated to the volume of 400 µL in phosphate buffered saline (PBS) and used for downstream analyses. These included nanoparticle tracking analysis (NTA) to determine EV concentration/yield, transmission electron microscopy (TEM) for EV size estimations and morphology assessment, microBCA protein quantification, liquid chromatography mass spectrometry (LCMS) for EV protein quantification, and nano flow cytometry for EV surface marker characterisation (EV-TRACK ID: EV210044).

**Figure 2 cells-10-00485-f002:**
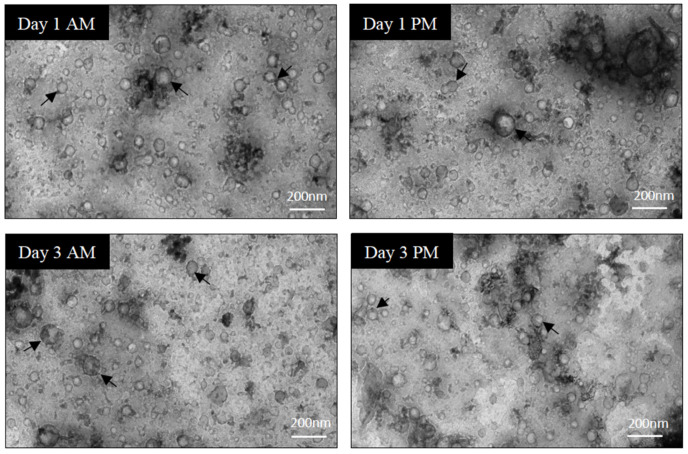
Sample quality assessment and characterisation of EVs using transmission electron microscopy (TEM). Direct mag: 30,000×, no sharpening, normal contrast. Scale bar = 200 nm.

**Figure 3 cells-10-00485-f003:**
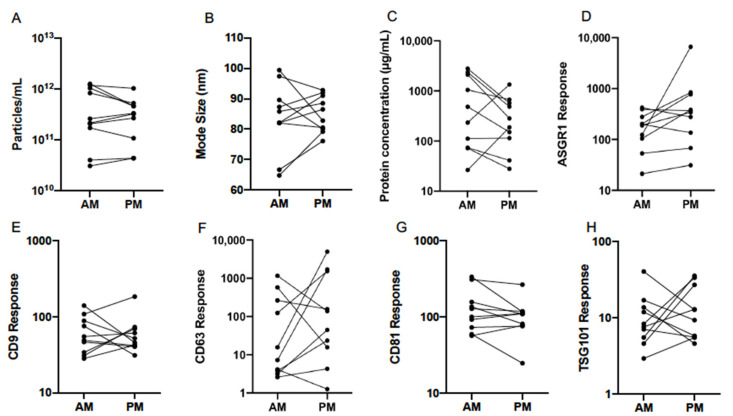
Variability of serum EV particle abundance (**A**), size (**B**), total protein concentration (**C**), and expression of EV-associated protein markers (**D**–**H**), quantified by nanoparticle tracking analysis (NTA), microBCA assay, and liquid chromatography mass spectrometry (LCMS) in the morning (AM) and afternoon (PM) of study day 1 in healthy volunteers (*n* = 10). ASGR1: asialoglycoprotein receptor 1; CD: cluster of differentiation; TSG101: tumour suppressor gene 101.

**Figure 4 cells-10-00485-f004:**
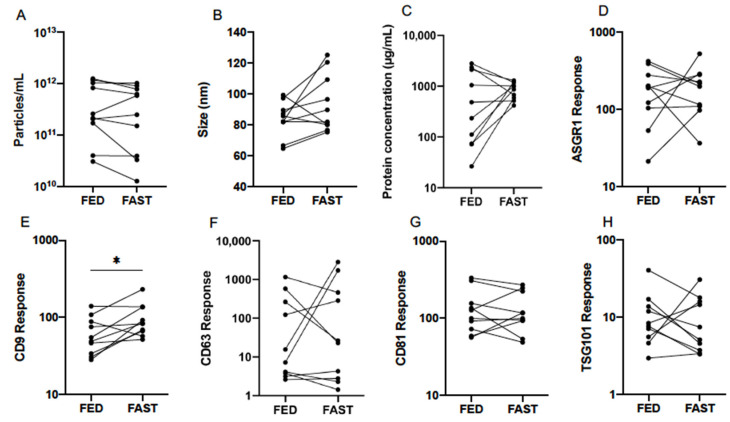
Effect of fed and fasted state on serum EV particle abundance (**A**), size (**B**), total protein concentration (**C**), and expression of EV-associated protein markers (**D**–**H**), quantified by nanoparticle tracking analysis (NTA), microBCA assay, and liquid chromatography mass spectrometry (LCMS) from healthy volunteers (*n* = 10). Statistical analysis performed using paired *t*-tests. ** p* ≤ 0.05.

**Figure 5 cells-10-00485-f005:**
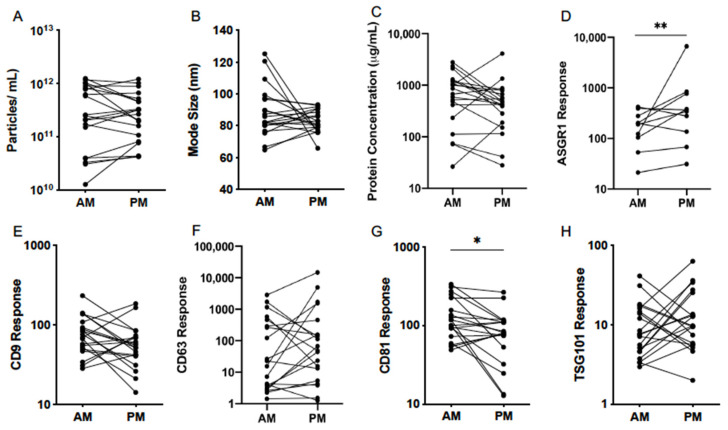
Diurnal variability of serum EV particle abundance (**A**), size (**B**), total protein concentration (**C**) and expression of EV-associated protein markers (**D**–**H**), quantified by nanoparticle tracking analysis (NTA), microBCA assay and liquid chromatography mass spectrometry (LCMS) in the morning (AM) and afternoon (PM) on study days 1 and 3 in healthy volunteers (*n* = 20). Statistical analysis performed using paired *t*-tests. ** p* ≤ 0.05, *** p* ≤ 0.01.

**Figure 6 cells-10-00485-f006:**
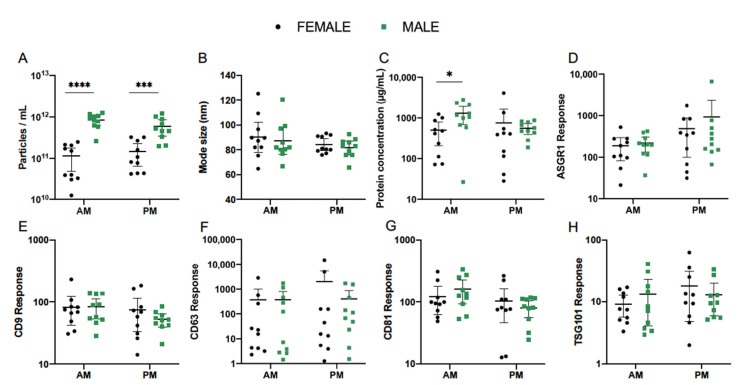
Effect of sex on serum EV particle abundance (**A**), size (**B**), total protein concentration (**C**), and expression of EV-associated protein markers (**D**–**H**), quantified by nanoparticle tracking analysis (NTA), microBCA assay, and liquid chromatography mass spectrometry (LCMS). Pooled analysis of EVs collected from serum of healthy female (*n* = 10) and male (*n* = 10) volunteers on study days 1 and 3. Statistical analysis performed using one-way ANOVA. ** p* ≤ 0.05, **** p* ≤ 0.001, **** *p* ≤ 0.0001.

**Table 1 cells-10-00485-t001:** Characteristics of EVV Study Cohort.

Characteristic	Healthy Females (*n* = 5)	Healthy Males (*n* = 5)
Age (years) Mean (Range)	28 (22–35)	30 (23–38)
Height (cm) Mean (±SD)	166.8 (5.7)	184.2 (8.0)
Weight (kg) Mean (±SD)	55.4 (3.8)	86.4 (5.4)
BMI (kg/m^2^) Mean (±SD)	20.0 (2.4)	25.5 (1.2)

**Table 2 cells-10-00485-t002:** Geometric mean, 95% confidence interval (CI), minimum and maximum of particle count, mode size, total protein concentration, and EV-associated protein marker abundance quantified in EVs isolated in the morning and afternoon on study day 1 from serum of healthy volunteers (*n* = 10).

	Particle Count (Particles/mL)		Mode Size (nm)	
	AM	PM	AM	PM
Mean	2.82 × 10^11^	2.40 × 10^11^	83.0	84.7
95% CI Lower	1.07 × 10^11^	1.12 × 10^11^	75.2	80.5
95% CI Upper	7.24 × 10^11^	5.13 × 10^11^	91.8	89.1
Minimum	3.02 × 10^10^	4.37 × 10^10^	64.9	76.0
Maximum	1.26 × 10^12^	1.02 × 10^12^	99.3	92.9
	**Protein Concentration (µg/mL)**		**ASGR1 Response**	
	**AM**	**PM**	**AM**	**PM**
Mean	347.5	219.8	147.9	331.9
95% CI Lower	104.2	90.8	76.6	115.9
95% CI Upper	1161.5	533.3	286.4	952.8
Minimum	26.4	27.9	21.2	31.2
Maximum	2799.0	1345.9	421.7	6622.2
	**CD9 Response**		**CD63 Response**	
	**AM**	**PM**	**AM**	**PM**
Mean	57.2	55.9	28.2	86.9
95% CI Lower	38.6	39.8	5.2	12.5
95% CI Upper	84.5	79.6	153.5	605.3
Minimum	28.2	31.0	2.6	1.3
Maximum	140.0	183.7	1158.8	4954.5
	**CD81 Response**		**TSG101 Response**	
	**AM**	**PM**	**AM**	**PM**
Mean	120.0	94.2	9.11	11.6
95% CI Lower	77.1	61.7	−0.73	6.6
95% CI Upper	187.1	143.9	15.5	20.4
Minimum	56.6	24.6	2.9	4.6
Maximum	334.2	264.9	40.9	35.8

**Table 3 cells-10-00485-t003:** Geometric mean and 95% confidence interval (CI) of particle count, mode size, total protein concentration, and EV-associated protein marker abundance quantified in EVs isolated from serum of healthy volunteers (*n* = 10) in fed state and after an overnight fast. Statistical analysis performed using paired *t*-tests. ns: not significant; * *p* ≤ 0.05.

	Particle Count (Particles/mL)				Mode Size (nm)			
	Fed	Fast	Difference	*p*	Fed	Fast	Difference	*p*
Mean	2.82 × 10^11^	2.09 × 10^11^	ns	0.193	83.0	92.0	ns	0.093
95% CI Lower	1.07 × 10^11^	6.61 × 10^10^			75.2	80.4		
95% CI Upper	7.24 × 10^11^	6.46 × 10^11^			91.8	105.4		
	**Protein Concentration (µg/mL)**				**ASGR1 Response**			
	**Fed**	**Fast**	**Difference**	***p***	**Fed**	**Fast**	**Difference**	***p***
Mean	347.5	822.2	ns	0.1602	147.9	168.7	ns	0.732
95% CI Lower	104.2	619.4			76.6	99.1		
95% CI Upper	1161.5	1094.0			286.4	287.7		
	**CD9 Response**				**CD63 Response**			
	**Fed**	**Fast**	**Difference**	***p***	**Fed**	**Fast**	**Difference**	***p***
Mean	57.2	90.6	*	0.018	28.2	41.5	ns	0.680
95% CI Lower	38.6	65.0			5.2	5.3		
95% CI Upper	84.5	126.2			153.5	322.9		
	**CD81 Response**				**TSG101 Response**			
	**Fed**	**Fast**	**Difference**	***p***	**Fed**	**Fast**	**Difference**	***p***
Mean	120.0	117.2	ns	0.886	9.1	7.9	ns	0.668
95% CI Lower	77.1	76.6			5.4	4.4		
95% CI Upper	187.1	179.5			15.5	14.1		

**Table 4 cells-10-00485-t004:** Pooled analysis of EVs collected from serum of healthy volunteers on study days 1 and 3 (*n* = 20) for comparison of morning and afternoon. Data presented as geometric mean and 95% confidence interval (CI) of particle count, mode size, total protein concentration, and EV-associated protein marker abundance. Statistical analysis performed using paired *t*-tests. ns: not significant, * *p* ≤ 0.05, ** *p* ≤ 0.01.

	Particle Count (Particles/mL)				Mode Size (nm)			
	AM	PM	Difference	*p*	AM	PM	Difference	*p*
Mean	2.29 × 10^11^	2.40 × 10^11^	ns	0.863	87.1	83.2	ns	0.240
95% CI Lower	1.23 × 10^11^	1.41 × 10^11^			81.3	77.6		
95% CI Upper	4.79 × 10^11^	3.89 × 10^11^			95.5	87.1		
	**Protein Concentration (µg/mL)**				**ASGR1 Response**			
	**AM**	**PM**	**Difference**	***p***	**AM**	**PM**	**Difference**	***p***
Mean	537.0	380.2	ns	0.123	147.9	331.1	**	0.009
95% CI Lower	295.1	223.9			97.7	169.8		
95% CI Upper	955	660.7			223.9	645.7		
	**CD9 Response**				**CD63 Response**			
	**AM**	**PM**	**Difference**	***p***	**AM**	**PM**	**Difference**	***p***
Mean	72.4	52.5	ns	0.075	33.8	67.6	ns	0.239
95% CI Lower	56.2	38.9			10.2	19.5		
95% CI Upper	93.3	70.8			114.8	234.4		
	**CD81 Response**				**TSG101 Response**			
	**AM**	**PM**	**Difference**	***p***	**AM**	**PM**	**Difference**	***p***
Mean	117.5	72.4	*	0.011	8.5	11.0	ns	0.293
95% CI Lower	49.0	12.9			5.9	7.6		
95% CI Upper	331.1	263.0			12.0	16.2		

**Table 5 cells-10-00485-t005:** Pooled analysis of EVs collected from serum of healthy female (*n* = 10) and male (*n* = 10) volunteers on study days 1 and 3 in the morning and afternoon. Data presented as geometric mean and 95% confidence interval (CI) of particle count, mode size, total protein concentration, and EV-associated protein marker abundance. Statistical analysis performed using one-way ANOVA. ns: not significant, ** p* ≤ 0.05, **** p* ≤ 0.001, **** *p* ≤ 0.0001.

	Particle Count (Particles/mL)							
	AM				PM			
	Female	Male	Difference	*p*	Female	Male	Difference	*p*
Mean	7.41 × 10^10^	7.76 × 10^11^	****	<0.0001	1.10 × 10^11^	5.01 × 10^11^	***	0.0002
95% CI Lower	3.47 × 10^10^	5.62 × 10^11^			6.03 × 10^10^	3.24 × 10^11^		
95% CI Upper	1.58 × 10^11^	1.10 × 10^12^			1.95 × 10^11^	7.94 × 10^11^		
	**Mode Size (nm)**							
	**AM**				**PM**			
	**Female**	**Male**	**Difference**	***p***	**Female**	**Male**	**Difference**	***p***
Mean	88.8	86.0	ns	0.610	83.9	81.3	ns	0.622
95% CI Lower	77.8	76.5			79.2	75.8		
95% CI Upper	101.4	96.8			88.8	87.2		
	**Protein Concentration (µg/mL)**							
	**AM**				**PM**			
	**Female**	**Male**	**Difference**	***p***	**Female**	**Male**	**Difference**	***p***
Mean	330.4	865.0	*	0.037	289.1	509.3	ns	0.822
95% CI Lower	154.5	335.0			97.5	358.1		
95% CI Upper	706.3	2238.7			855.1	722.8		
	**ASGR1 Response**							
	**AM**				**PM**			
	**Female**	**Male**	**Difference**	***p***	**Female**	**Male**	**Difference**	***p***
Mean	136.5	183.2	ns	0.801	248.9	321.4	ns	0.844
95% CI Lower	70.2	110.4			94.6	129.1		
95% CI Upper	265.5	304.1			654.6	799.8		
	**CD9 Response**							
	**AM**				**PM**			
	**Female**	**Male**	**Difference**	***p***	**Female**	**Male**	**Difference**	***p***
Mean	70.6	73.5	ns	0.987	56.8	49.3	ns	0.837
95% CI Lower	46.7	49.8			32.1	37.8		
95% CI Upper	106.9	108.4			100.7	64.4		
	**CD63 Response**							
	**AM**				**PM**			
	**Female**	**Male**	**Difference**	***p***	**Female**	**Male**	**Difference**	***p***
Mean	29.6	39.5	ns	0.964	57.9	78.9	ns	0.959
95% CI Lower	5.1	5.3			6.5	15.2		
95% CI Upper	171.0	293.1			515.2	408.3		
	**CD81 Response**							
	**AM**				**PM**			
	**Female**	**Male**	**Difference**	***p***	**Female**	**Male**	**Difference**	***p***
Mean	103.50	135.8	ns	0.641	73.0	71.9	ns	0.999
95% CI Lower	69.2	87.5			35.4	48.4		
95% CI Upper	155.2	210.9			150.3	107.2		
	**TSG101 Response**							
	**AM**				**PM**			
	**Female**	**Male**	**Difference**	***p***	**Female**	**Male**	**Difference**	***p***
Mean	8.0	9.0	ns	0.945	11.7	10.5	ns	0.944
95% CI Lower	5.3	4.6			5.7	6.6		
95% CI Upper	12.1	17.6			24.2	16.7		

## Data Availability

All data are included in the paper or attached as Appendix A.

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
