# Peer review of "Importance of between and within Subject Variability in Extracellular Vesicle Abundance and Cargo when Performing Biomarker Analyses"

_cells, 2021, doi:10.3390/cells10030485_

Round 1

Reviewer 1 Report

Newman et.al., demonstrate that EV abundance and cargo was generally not affected by fasting, except CD9 which exhibited a statistically significant increase. Moreover, they also show that diurnal variability was observed in the expression of CD81 and ASGR1 which significantly decreased and increased, respectively.

These results are very valuable because they are basic studies that lead to the understanding of liver health and the isolation of effective biomarkers such as drug metabolism.

However, in this paper, there are the following problems.

1  In Table 1, it is difficult to analyze the effect of sex without describing the characteristics by gender.

2  In this study, blood was collected in the morning and afternoon in the same person, so more useful data can be obtained by connecting the morning and afternoon values of the same person in all figures. It is possible that some people have large fluctuations in the value, and many have not changed much.

3  Regarding fasting, are the subjects provided with the same meal content? Please describe the information.

4  As for gender differences, what is reflected in the discussion.

5  In this study, exosomes are isolated using the qEV70 2mL SEC column. Please explain the reason for the selection.

Reviewer 2 Report

The authors sought to evaluate variability in circulating global and liver-specific sEV abundance and cargo, and to define patterns of variability, within (diurnal) and between individuals, that have the potential to confound sEV derived biomarker analyses. 

The experimental design and results look good. 

Author Response

We thank the Reviewer for the comments and the time considering our manuscript.